# Learning to Segment the Lung Volume from CT scans based on Semi-Automatic Ground-Truth

**Patrick Sousa**
INESC TEC
Porto, Portugal
patrick.sousa@inesctec.pt

**Adrian Galdran**
INESC TEC
Porto, Portugal
adrian.galdran@inesctec.pt

**Pedro Costa**
INESC TEC
Porto, Portugal
pvcosta@inesctec.pt

**Aurélio Campilho**
INESC TEC
Faculdade de Engenharia da
Universidade do Porto
Porto, Portugal
campilho@fe.up.pt

## Abstract

Lung volume segmentation is a relevant task within the design of Computer-Aided Diagnosis systems related to automated lung pathology analysis. Isolating the lung from CT volumes can be a challenging process due to the considerable deformations and pathologies that can appear in different scans. Deep neural networks can be an effective mechanism in order to model the spatial relationship between different lung voxels. Unfortunately, this kind of models typically require large quantities of annotated data, and manually delineating the lung from volumetric CT scans can be a cumbersome process. In this paper, we propose to train a 3D Convolutional Neural Network to solve this task based on semi-automatically generated annotations. To achieve this goal, we introduce an extension of the well-known V-Net architecture that can handle higher-dimensional input data. Even if the training set labels are noisy and may contain some errors, we experimentally show that it is possible to learn to accurately segment the lung relying on them. Numerical comparisons performed on an external test set containing lung segmentations provided by a medical expert demonstrate that the proposed model generalizes well to new data, reaching an average 98.7% Dice coefficient. In addition, the proposed approach results in a superior performance when compared to the standard V-Net model, particularly on the lung boundary, achieving a 0.576 mm Average Symmetric Surface Distance with respect to expert validated ground-truth.

## 1 Introduction

In Computer-aided diagnosis (CAD) of pulmonary diseases, lung volume segmentation is an essential preliminary pre-processing stage intended to isolate the lung from the background. Accurate lung segmentation is of great importance, since it allows to avoid unnecessarily processing irrelevant information and it enables false positive removal, thereby preventing potentially incorrect diagnosis.

Several automated methods for lung segmentation have been developed along the years, specially on Computer Tomography (CT) images. Most of these methods are threshold [7] or region-based [15], relying mostly on intensity levels, contrast and neighborhood homogeneity. More sophisticated methods are based on prior anatomical knowledge. This is the case of atlas-based methods, which rely on the registration of the target image to a template image containing labels of the thoracic region

1st Conference on Medical Imaging with Deep Learning (MIDL 2018), Amsterdam, The Netherlands.

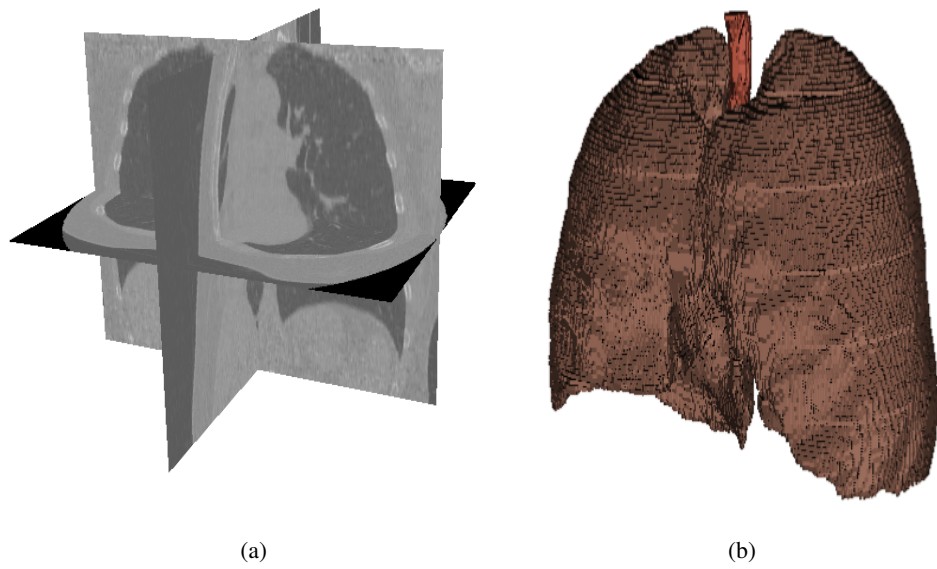

(a)                                                       (b)

Figure 1: (a) Example of a volumetric CT scan of the lung extracted from the dataset provided in the LUNA16 competition [8] (b) Automatically generated volumetric segmentation of the lung.

[10]. Neighboring anatomy-guided methods use spatial information about the surrounding organs to delineate the lung regions, in order to simplify the segmentation task in cases where abnormalities or artifacts are present. Hybrid approaches combining fast traditional threshold-based techniques with more sophisticated multi-atlas methods have also been proposed [14]. In this case, a segmentation obtained using conventional approaches is automatically examined for errors, and corrected by means of more time-consuming atlas-based methods.

A wide variety of techniques has also been proposed for the task of segmenting 3D volumetric organs from CT scans. Dou *et al.* [1] proposed a 3D deeply supervised model based on a Fully Convolutional Network (FCN) to automatically segment the liver on CT images. In order to segment multiple organs in a CT scan, Roth *et al.* [13] adapted an existent architecture called 3D U-Net [16]. For lung segmentation in CT scans, Harrison *et al.* [4] introduced a deep architecture termed Holistically-nested network (HNN). This model was particularly accurate at finely delineating lung borders. In addition, a progressive multi-path scheme was also implemented in order to deal with issues related to output ambiguity and coarsening resolution, resulting in an extended method called Progressive Holistically-nested network (P-HNN).

In general, deep neural networks are known to depend on the availability of large quantities of annotated data. Unfortunately, for the problem of lung segmentation, there exist few public sources of such data. On the other hand, semi-automatic segmentations of the lung in CT scans can be easily generated. In the LUng Nodule Analysis 2016 (LUNA16) challenge [8], a competition aimed at automatic lung nodule detection, such ground-truth was provided by the challenge based on the CT scans from the Lung Image Database Consortium (LIDC) and Image Database Resource Initiative (IDRI). In this case, lung segmentations were generated by a semi-automatic method [14], resulting in reasonably accurate annotations, see Figure 1. However, it is important to stress that these annotations are not perfect, and were not validated by a medical doctor. Therefore, they should not be used for clinical evaluation purposes.

Noisy ground-truth and pseudo-labels have recently proven useful for training deep learning-based segmentation models on brain MRI images in [3], where it was shown that models trained on this kind of imperfect annotations can generalize properly to new data and achieve great performance when evaluated with clinically correct ground-truth. Following this approach, in this paper we propose to use the noisy automatically generated ground-truth provided from the LUNA16 challenge

to train an extension of the V-Net architecture [11] for the task of lung segmentation. Our main contribution is the demonstration that this kind of pseudo-annotations are indeed useful for this task. We experimentally verify our approach by testing the model on a separate dataset containing lung volumes that were manually delineated by a clinical expert. Experimental results demonstrate that our model, even if trained on semi-automatically generated labels, is effective at segmenting the lungs from CT scans when tested on ground-truth provided by a medical doctor.

## 2 Methodology

### 2.1 Model Architecture

One of the most popular methods for medical image segmentation is the U-Net [12] model, a deep Convolutional Neural Network (CNN) architecture that was first proposed for the task of segmenting neuronal structures in electron microscopic stacks. This architecture is an extension of Fully-Convolutional Networks consisting of a downsampling followed by an upsampling path. In the downsampling path, similarly to a standard CNN architecture, the input passes through several layers of convolutional blocks of stride two (which subsequently reduce the spatial resolution of the output volumes) followed by Rectified Linear Unit (ReLU) activations. Along the upsampling path, for each layer the result of the previous layer is concatenated with the one of the corresponding layer from the downsampling path, which is passed trough skip connections, thereby allowing to avoid information loss from higher scales of the image.

The U-Net architecture has been later generalized by several authors to handle three-dimensional data. For instance, in [16] 3D U-Net was introduced, while V-Net was presented [11]. In both cases, the spatial resolution of the input images was relatively limited: 3D U-Net was applied on 3D confocal microscopy images of the Xenopus kidney, training with volumes of approximately $245 \times 244 \times 56$ resolution, while the V-Net model was proposed to segment the prostate in 3D MRI images dealing with an input size of $128 \times 128 \times 64$.

This reduced input spatial resolution is related to memory constraints arising when dealing with three-dimensional medical data. When the available data has a much larger spatial resolution than the size of the input in a previously defined architecture like 3D U-Net or V-Net, a solution needs to be implemented to deal with such memory constraints. A possible approach is to resize the spatial dimensions of the data to the size of the input of the architecture. This is a simple and fast strategy, but in some cases it can lead to a critical loss of relevant information for the task to be solved. An alternative approach consists of dividing the input image into volumetric patches of the size of the desired network's input. The output of the model is then a spatial reconstruction of several outputs, corresponding to the initial division of the input volume. With this procedure, small details are not lost, but certain contextual information is missed at the boundaries of the divided sub-volumes.

In this paper, for lung segmentation from CT scans we prefer to consider a spatial resolution substantially greater than those employed in previous 3D deep CNN models. Therefore, to deal with large data dimensionality while avoiding information loss, we propose a novel strategy. Specifically, starting from the initial V-Net architecture, we introduce a max-pooling layer early in the model in order to reduce its dimensionality from $512 \times 512 \times 256$ to the conventional $128 \times 128 \times 64$ of V-Net. Then, to mitigate the information lost in this operation, we introduce a skip connection between the input of our architecture and the last convolutional layer. We also introduce another relevant modification to the original V-Net design, namely we reduce the number of filters in our model to $2/3$ of the ones used in the original architecture. We experimentally verified that the resulting model is less prone to overfitting the training data, while still producing highly accurate predictions. Finally, ReLU non-linearities were replaced by PReLu [5] activation functions, and batch normalization was also added in our architecture. An overall diagram of the proposed architectural design is represented in Figure 2.

### 2.1.1 Loss Function

Similarly to the original V-Net architecture, the loss function minimized during training the proposed model is based on the Dice coefficient. The Dice coefficient is a classical segmentation metric useful for measuring the overlapping volume between two different three-dimensional objects. In this case,

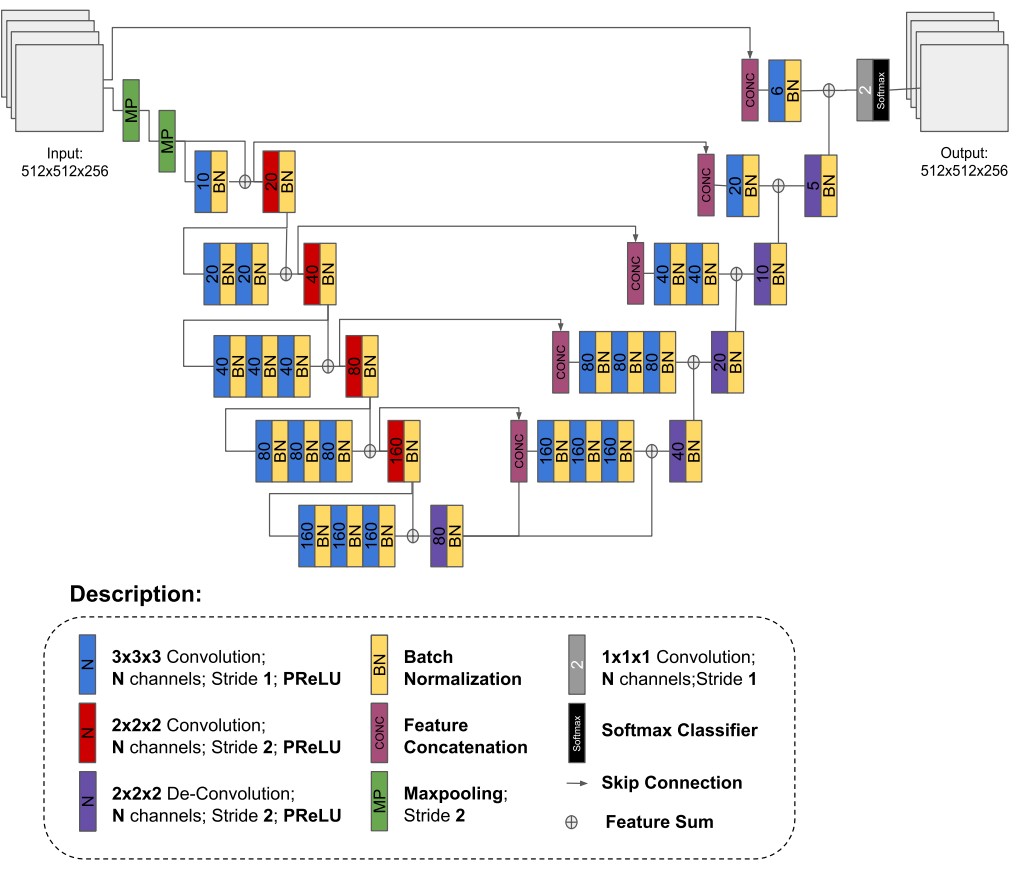

Figure 2: The proposed model, based on the V-Net, with the addition of an extra initial layer and skip connections allowing to deal with $512 \times 512 \times 256$ size input CT scans.

it is embedded in the loss function $D(P, G)$, which can be written as:

$$D(P, G) = \frac{2 \sum_{i}^{N} p_i g_i}{\sum_{i}^{N} p_i^2 + \sum_{i}^{N} g_i^2},$$

where $N$ is the number of voxels on each image, $p_i \in P$ represents a voxel $p_i$ within the predicted segmentation $P$, and $g_i \in G$ is the binary ground-truth segmentation.

## 2.2 Training Set and Semi-Automatically Generated Ground-Truth

The data used for training the above model was provided by the LUNA16 challenge [8], composed of 888 CT scans extracted from the Lung Image Database Consortium (LIDC) and Image Database Resource Initiative (IDRI) [6]. LUNA16 is accompanied with volumetric lung segmentations for each scan, distinguishing from left and right lung. For this work, we merged the labels from each lung into a single ground-truth volume, resulting in data and labels similar to those depicted in Figure 3.

The lung segmentations released in [8] were generated automatically, and may contain certain amount of errors. Accordingly, these annotations should not be used as a reference in any segmentation study. It is important to stress that these segmentations are not used for testing our algorithm, but only to train the model. The main hypothesis we aim to verify is that a deep CNN can be trained on such imperfect noisy ground-truth and still learn useful representations. The model is thus trained to generate lung segmentations that are to be validated in test time with a separate dataset of manually delineated lung volumes, see Section 3.1.

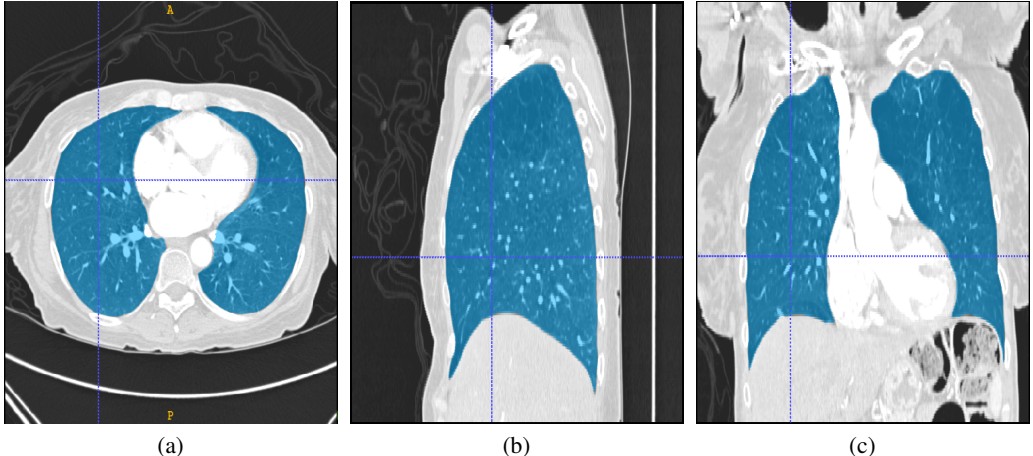

|         (a)         |         (b)         |         (c)         |

Figure 3: Axial, Sagittal and Coronal CT scan views of a lung volume from the LUNA16 dataset and the correspondent ground-truth employed in this work.

For training, the available data was divided into 700 scans for training and 188 for validation that were randomly selected. The scans have a fixed spatial resolution of $512$ and a slice thickness ranging between 0.6mm to 2.5mm. Before being supplied to the proposed model, the depth resolution of the scans was mapped to a common value of 256 voxels. In addition, it is well-known that in CT scans the majority of the relevant information lies in the Houndsfield units range of $[-1000; 400]$. Accordingly, information outside this range was omitted. After the pre-processing stage, the images are normalized to zero mean and unit variance. Standard data augmentation techniques (spatial shifting, zooming along the depth axis) are also applied to increase the training data.

## 2.3   Implementation and Training

The model was trained with standard backpropagation for 12 epochs using the Adam Optimizer [9] and an initial learning rate of $1e^{-3}$. On each epoch, the model runs through all the training set, with each batch constituted by a single scan due to memory constraints. The loss defined in eq. (1) was also monitored in the validation set, and training was early-stopped when the validation loss was not improving for a pre-determined number of epochs. The proposed method was implemented in Python 2.7 using the PyTorch framework. The workstation has an Intel Xeon E5-2630 v4 CPU at 2.20GHz, 31 Gb of RAM and a NVIDIA Titan P100 with 16 Gb GPU.

# 3   Experimental Setting

## 3.1   Test Set

The proposed method was evaluated using data from the VESSEL12 Challenge that provided 20 CT scans of the chest. These scans are of size $512 \times 512$, with a variable depth resolution of a maximum spacing of 1mm. The dataset contained both healthy and pathological lungs. The lung volume ground-truth data was acquired and validated by an expert radiologist. It contains labels dividing the lung volume into its different lobes in order to train lobe segmentation models [2]. In this case, since we are only interested in the overall lung region, we merge the annotations from all lobes into a single label. In Figure 4, a CT scan with its correspondent volumetric lung ground-truth extracted from the VESSEL12 dataset is displayed.

## 3.2   Experimental Evaluation

To evaluate our model, the Dice coefficient was used. As observed in section 2.1.1, this metric returns the value of the overlapping volume between the ground-truth and the binarized predicted segmentation. Since it is an overlapping metric, when comparing very large objects errors present in

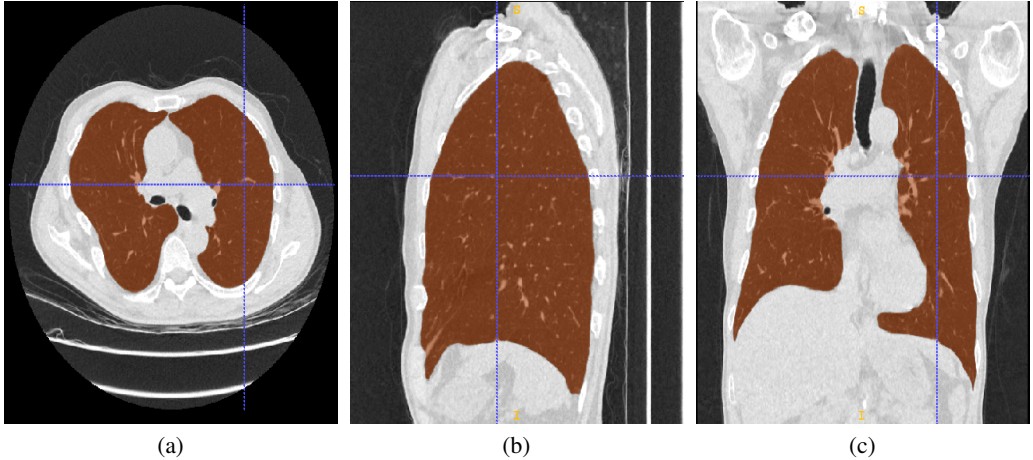

(a)                (b)                (c)

Figure 4: a) Axial, b) Sagittal, and c) Coronal views of a CT scan from the lung, extracted from the the VESSEL12 dataset, with corresponding lung segmentations.

Table 1: Comparison of the Dice Coefficient and Average Symmetric Surface Distance (ASD) of the results from the proposed model and V-Net.

|  | Dice Coefficient (%) | Average Surface Distance (mm) |
|---|---|---|
| V-Net | 97.2 | 2.627 |
| Ours | **98.7** | **0.576** |

the border of the prediction will not affect the value of the Dice coefficient. As such, and considering also that the Dice coefficient is also embedded in the loss function we are minimizing, in order to perform a more fair experimental evaluation the predicted segmentations were also assessed by the Average Symmetric Surface Distance (ASD). ASD is a surface distance metric that measures the average distance of all the points of the surface of the 3D segmentation with their closest points in the surface associated to the ground-truth.

$$ASD = \frac{1}{|B_{seg}| + |B_{gt}|} \cdot \left( \sum_{x \in B_{seg}} d(x, B_{gt}) + \sum_{x \in B_{gt}} d(y, B_{seg}) \right) \tag{2}$$

## 4  Results and Discussion

In order to evaluate our model, after training the model on the training data and semi-automatically generated ground-truth, we produced predictions for the 20 scans on our independent test set, and compared them with the corresponding manually generated ground-truth. In Figure 5, we show examples of the predictions generated by our method, which in this case produces a fine lung volume segmentation.

For comparison purposes, we trained a standard V-Net on the same dataset. The model received as input the downsampled CT scan to $128 \times 128 \times 64$. In Figure 6 we present a prediction of the same CT scan shown in Figure 5. In this case, the depicted segmentation was generated by the V-Net. A more detailed visual comparison is provided in Figure 7. As can be observed, the volume in general is well predicted by both models, although the approach introduced in this paper achieves a more finely delineated boundary, while in the segmentation produced by the V-Net model, the precision on the boundaries is affected by the downsampling and upsampling processes, resulting in small stair-casing effects along the lung borders. Nevertheless, both results are relatively satisfactory, verifying this way that a deep 3D CNN can be effectively trained on the kind of noisy ground-truth used in this work.

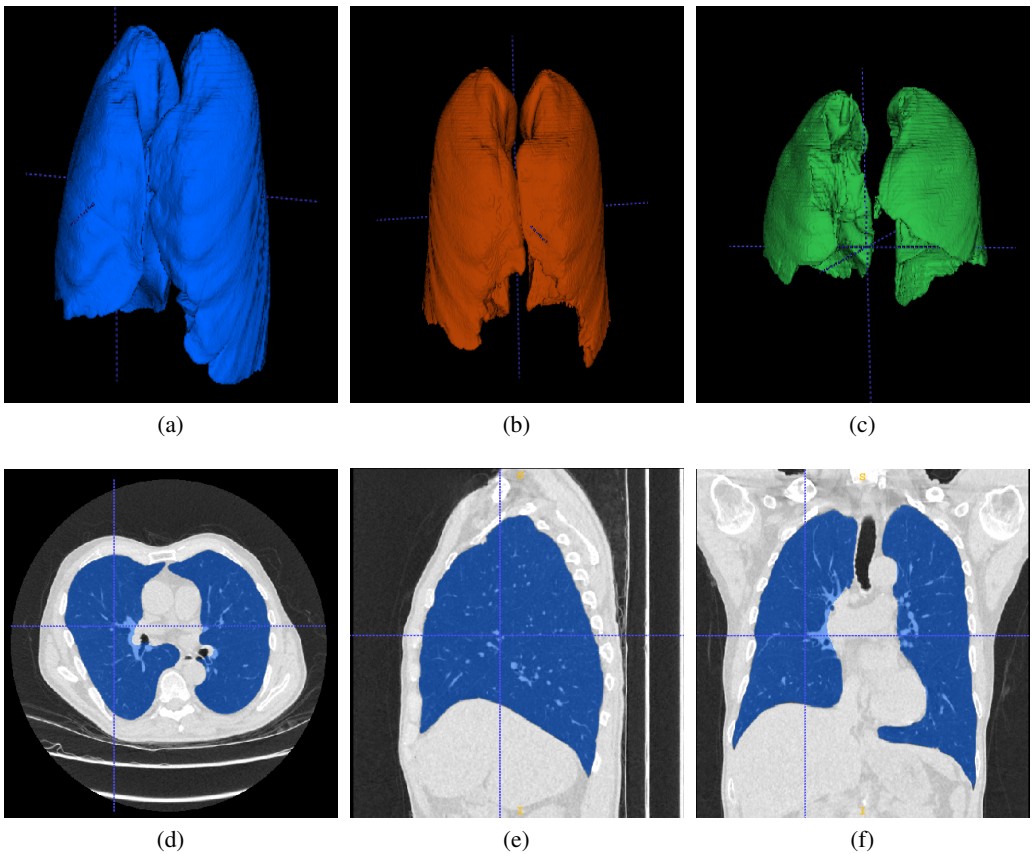

Figure 5: a), b), and c): 3D segmentations produced by our model from CT scans belonging to the test set. d), c), and f): Axial, Sagittal and Coronal views of the lung segmentation shown in a).

Finally, for a numerical comparison of both methods, we computed Dice coefficients and ASM over the entire available test set. The results are reported in Table 1. We can observe that both methods achieve a good Dice coefficient score in the manually-annotated test set, numerically demonstrating that both models learned to properly segment the lung volume, even when trained on imperfect ground-truth of our train set. Furthermore, the proposed model is shown to achieve a slightly better Dice coefficient than the standard V-Net. This is further verified by observing the ASD values obtained by each model. The ASD achieved by our proposed extension to V-Net seems to be capable of better handling lung surface voxels, resulting in better segmented boundaries.

## 5   Conclusions and Future Work

In this paper, we have demonstrated that modern 3D segmentation methods based on Deep Convolutional Neural Networks can be effectively trained on imperfect automatically generated ground-truth for the task of lung volume segmentation from CT scans. In addition, we introduced an extension of the well-known V-Net architecture that can handle better surface voxels inside the lung. The proposed model can be supplied with scans of a $512 \times 512 \times 256$ resolution, thereby avoiding any initial information loss, and properly dealing with memory constrains. The proposed model produces highly accurate lung volume segmentations when validated in an external test set containing ground-truth provided by a medical expert, achieving a Dice Coefficient of 98.7% and an Average Surface Distance of 0.576mm. which are superior to results produced by a standard V-Net.

In future work, we will explore certain modification to the loss function driving the optimization process, in order to dedicate more attention to boundary errors. Another interesting research direction is the potential extension of the segmentation method to other pulmonary regions for which manual

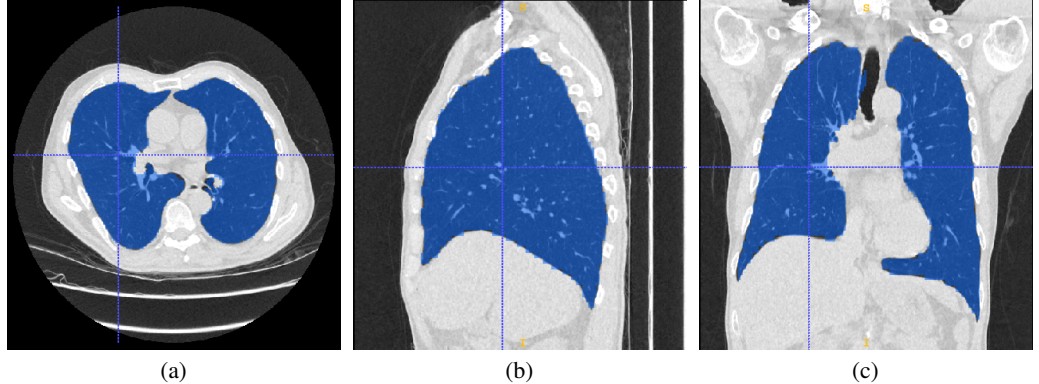

(a)                          (b)                          (c)

Figure 6: a) Axial, b) Sagittal, and c) Coronal views of the Segmentation from a CT scan produced by a standard V-Net model.

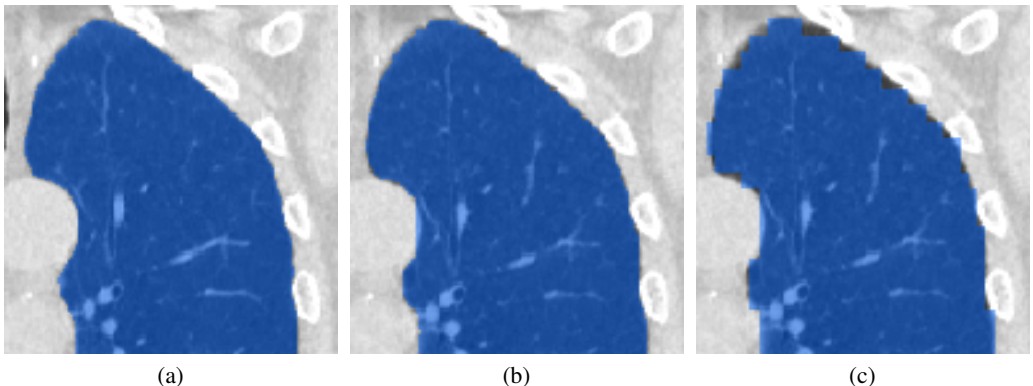

(a)                          (b)                          (c)

Figure 7: Visual comparison between a) the considered manual ground-truth, b) the segmentation produced by the proposed model, and c) the segmentation generated by a standard V-Net model.

ground-truth is hard to acquire, based on automatically generated segmentations that may be used for training such models.

## Acknowledgments

This work is funded by the North Portugal Regional Operational Programme (NORTE 2020), under the PORTUGAL 2020 Partnership Agreement, and the European Regional Development Fund (ERDF), within the project "NanoSTIMA: Macro-to-Nano Human Sensing: Towards Integrated Multimodal Health Monitoring and Analytics/NORTE-01-0145-FEDER-000016".

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
