# OpenReview forum: "Learning to Segment the Lung Volume from CT scans based on Semi-Automatic Ground-Truth"
_MIDL.amsterdam/2018/Conference — Submitted to MIDL 2018_

### Review · AnonReviewer1 · 2018-05-04
**simple pipeline reducing manual labor of getting annotations, minor architectural improvement, missing comparisons**

**Rating:** 2
**Confidence:** 2

**Review:**

This paper proposes to segment lung volume from CT scans using noisy labeled data. To do so, they augment the vnet architecture with a couple of additional max pooling, and a skip connection from input to the pre-segmentation layer.

pros
+ paper well written and easy to follow
+ simple pipeline that reduces the manual labor of getting voxel-wise annotations

cons
- the extension of vnet consists on a minor improvement of hyper-parameter search (better adapting the architecture to the problem)
- experimental results are not compared to the standard scenario where accurate labels are available

My main concerns are related to the contribution and the experimental evaluation.

Vnet (and many other 3D U-net and FCN variants) have been largely explored in the medical imaging literature to segment different kinds of organs and lesions, in different modalities. Changing ReLU/PReLU, adding a couple of max pooling layers and a skip connection to the original vnet seems a rather minor change. It is worth noting, that vnet was introduced to tackle a different problem, and thus applying it out of the box may not lead to the best results, requiring proper architectural hyper-parameter search to be performed.

In order to highlight the importance of the changes introduced, it might be relevant to analyze how each one of these changes affects the performance of the model.

Moreover, the choice of adding two max pooling layers seems rather arbitrary, why not add a single max pooling with a larger kernel and stride?

The paper claims that the second contribution is to show how using noisy labels (from a segmentation method) is good enough to train their model. Although the use of noisy labels is an interesting line of research, it seems that this ideas have already been explored in medical imaging with brain MRIs. This work is properly cited in the paper [3], but it is not clear how the proposed approach differs from the previous one.

In the experimental section, it seems that noisy labels are used for the validation set. Why not use a clean validation set?

To understand the advantages and limitations of using noisy labels, a comparison with the model trained on non-noisy labels would be beneficial.

Additional comparisons to other models in the literature would make the results more compelling, so far the only comparison performed is with the out-of-the-box vnet.

It seems that the model was only trained for 12 epochs and with early stopping, what was the patience?

In Eq (2), some notation is not introduced. I think the second sum should be on y not x. How is d defined?

Since ASD is used to evaluate the model, would it be reasonable to include it in the loss function?

Figure 5 & 6 could be merged, to ease comparison.

Some references are missing:
* reference to Pytorch
* reference to [a,b] for dice loss

[a] https://arxiv.org/pdf/1608.04117.pdf
[b] https://arxiv.org/pdf/1606.04797.pdf

**Special Issue:**

No

---

### Review · AnonReviewer2 · 2018-05-07
**Nice manuscript, limited novelty, good results**

**Rating:** 3
**Confidence:** 2

**Review:**

The authors present a V-Net based method for segmenting the lungs in lung CT scans. Overall I think this is a good paper but the novelty is limited. I did like the impact the changes to the VNet architecture seemed to have on algorithm performance. What I was missing was any kind of analysis of what changes to the architecture had what impact. I suspect a large part of the lower performance of the standard V-net had to do with the subsampling. It would have been nice had the authors elaborated and done more experiments to see what impact some of the other changes they made had on the result. Finally, given that there are both diseased and non-diseased scans in the test set I was expecting there to be an analysis of performance on both types of scans separately. In general I think some analysis of where the proposed method fails would be interesting. There are no examples of failed cases and thus no discussion of concrete issues with the method (maybe there are none?).

Pros:
- Authors use public data
- Results seem good
- Architecture and training method described in reasonable detail

Cons:
- Limited discussion and no analysis of diseased vs non-diseased scan results.

Detailed review:

I only have a couple of comments:

2.3 implementation and training: I am unfamiliar with the "NVIDIA Titan P100 with 16GB of RAM". Are the authors referring to the NVIDIA Quadro P100?

3.1 Test set: how did the expert radiologist acquire the reference standard from the images?




**Special Issue:**

No

---

### Review · AnonReviewer3 · 2018-05-08
**The paper describes an experiment aiming to prove that an FCN model can be trained successfully on noisy labels. For that, authors train FCN models for lung segmentation using lung labels generated by a semi-automatic method and evaluate on lung segmentations validated by an expert radiologist. Additionally, a modified version of the V-Net architecture is proposed, which given the same memory constraints can be trained on images with 4x bigger spatial resolution than the original V-Net.**

**Rating:** 2
**Confidence:** 2

**Review:**

Pros
+ The paper addresses a problem of learning on noisy labels which is relevant in the medical field, because acquisition of high quality annotations is very costly.
+ The datasets used for training and evaluation (LUNA16, VESSEL12) are publicly available, which makes it easier to reproduce the work.
+ The paper reads well and is well structured.

Cons
- The main hypothesis of the paper stating that models trained on noisy labels can learn good representation and thus deliver accurate results is not well proved, because it is not clear how “noisy” the training labels are. For that, the quality of the semi-automatic method used to generate the training labels should also be evaluated on the test data.
- The authors describe an approach of dealing with GPU memory constraints by dividing input images into volumetric patches which fit on a GPU. They claim that with this approach the trained network misses contextual information at the boundaries of the divided sub-volumes, which is not true if convolutions in “valid” mode are employed.
- The comparison of the proposed model and the original V-Net using the ASD metric is in my opinion inconclusive, because the V-Net model was trained with images downsampled by a factor of four in comparison to the proposed model. To allow for a fair comparison, both models should be trained with images in the same resolution (it would be possible using the overlap-tile approach described in the 3D U-Net paper, which they cited).
- The authors claim that reduction of filters to 2/3 of the ones used in the original make the model less prone to overfitting without affecting the model accuracy. There are no results presented in the paper which support this statement.

Other comments
- Fig 2.: Most of the feature concatenation layers have only one incoming connection (a connection from a lower level is missing?). There is channel count mismatch at the first (1 and 10 channels) and last (6 and 5) feature sum operation, which in my opinion requires additional explanation.
- It would be beneficial to report inter-observer variability of radiological experts to put the results of automatic methods into a perspective.
- Sec. 4, first sentence of the last paragraph: Authors refer to an ASM metric, which is not introduced before. I think, that the ASD metric was meant.


**Special Issue:**

No

---

### Decision · Program_Chairs · 2018-05-15
**Paper37 Acceptance Decision**

Reject